# Is It Necessary to Centralize Power in the CEO to Ensure Environmental Innovation?

**Beatriz Aibar-Guzmán** [1] and **José-Valeriano Frías-Aceituno** [2,*]

1    Facultad de Ciencias Económicas y Empresariales, University of Santiago de Compostela, 15782 Santiago de Compostela, Spain; beatriz.aibar@usc.es
2    Facultad de Ciencias Económicas y Empresariales, University of Granada, 18071 Granada, Spain
*    Correspondence: jfrias@ugr.es

**Abstract:** Using data from a sample of 4863 international firms corresponding to the period 2002–2017, this paper examines the role that chief executive officer (CEO) power plays in environmental innovation and the impact that these strategies have on financial performance. Both issues have been the subject of considerable debate in the literature, with opposite views and contradictory findings. The results indicate that investing in environmental innovations related to the use of clean technologies, ecological production processes, and the design, manufacture and commercialization of environmentally sustainable products requires that CEOs have a greater degree of power in order to support projects that do not entail a higher return in the short and medium terms. Additionally, the results show that the negative economic effect of eco-innovation reverses in the fourth and fifth years after environmental innovations were implemented. Thus, this study supports the view regarding a "bright side" of CEO power with regard to corporate sustainability.

**Keywords:** eco-innovation; environmental innovation; CEO power; CEO ability; financial performance; upper echelon theory





## 1. Introduction

As a result of increasing stakeholder demands for corporate environmentally responsible behavior, in recent decades eco-innovation has become increasingly important for companies worldwide (Amore and Bennedsen 2016; Huang et al. 2021) as an essential way to achieve environmental sustainability and fulfill their social responsibility (Guoyou et al. 2013; Liao et al. 2019; García-Sánchez et al. 2020a). In parallel, eco-innovation has attracted growing research attention (de Jesus Pacheco et al. 2018) with the aim of knowing its characteristics and drivers (Díaz-García et al. 2015). However, most studies have focused on examining the institutional and market factors as well as the firm characteristics that lead companies to pursue eco-innovation (Keshminder and Río 2019), whereas researchers have paid relatively little attention to the role played by the chief executive officer (CEO) in this regard (Liao et al. 2019), despite being the main actor responsible for corporate strategies, including those related to corporate social responsibility (CSR) (Arena et al. 2018; García-Sánchez and Martínez-Ferrero 2019; García-Sánchez et al. 2020c). Indeed, only a small number of papers have analyzed how the CEO's characteristics may affect her/his company's propensity to develop eco-innovation.

According to Sheikh (2018, p. 36), CEOs are "the chief planners and architects of a firm's innovation strategy". Furthermore, prior research has showed that management support plays a key role in the generation and implementation of environmental innovations (Daily and Huang 2001; Qi et al. 2010; Agan et al. 2013; Hojnik and Ruzzier 2016). As eco-innovation projects are characterized by high information asymmetries (Demirel and Parris 2015) and managerial discretion (Oh et al. 2016), they will be strongly influenced by the CEO's preferences and priorities which, in turn, are affected by her/his individual

characteristics, skills, and values (Arena et al. 2018; Liao et al. 2019; Wu et al. 2020; Huang et al. 2021).

This can be explained from the upper echelon theory's perspective (Hambrick and Mason 1984), which posits that CEO idiosyncratic differences (e.g., gender, age, background, ideology, hubris, religious beliefs, managerial ability, etc.) influence corporate strategic decisions and outcomes (Hambrick 2007). Such idiosyncratic characteristics also affect the CEO's perception about the salience of sustainability problems (Lewis et al. 2014; Arena et al. 2018) and her/his sensibility to them, so that not all CEOs are equally aware that global warming and the deterioration of the natural environment represent a serious problem that affects the security of people and are willing to take actions to reduce the environmental impacts of their companies' activities. Thus, some CEOs strongly believe that global warming and environmental problems are very important and look for initiatives to advance towards a sustainable economy, whereas, conversely, other CEOs believe that such problems are not significant or will "resolve themselves" and, consequently, they remain uncommitted in relation to environmental protection.

However, given that the degree of CEO's power and influence may vary (Sheikh 2018), it is logical to assume that the CEO's influence on her/his company's propensity to develop eco-innovation may be different. Thus, powerful CEOs have more room to make unilateral decisions by imposing their judgment (Eisenhardt and Bourgeois 1988; Haleblian and Finkelstein 1993). On the contrary, the less powerful CEOs' decisions tend to be conditioned by the opinion and approval of the supervisory bodies of the work of the CEO and her/his team, such as the board of directors.

With these premises, and assuming that the CEO's perception of the significance of sustainability problems leads her/him to held a positive attitude towards eco-innovation, this paper aims (1) to determine the extent to which the accumulation of power in the figure of the CEO can be determinant of the companies' decision to eco-innovate, and (2) to analyze the economic effect of these decisions. In other words, this study is driven by the following research questions: (1) How does CEO power affect eco-innovation? and (2) How does eco-innovation affect firm performance? Both issues have been the subject of controversy among scholars and there is a lack of consensus on the sign of such relationships. Hence, answering these questions, we aim to contribute to the academic debate surrounding the drivers and outcomes of eco-innovation.

The empirical evidence obtained for a sample of 4863 international companies for the period 2002–2017 shows that investing in environmental innovations related to the use of clean technologies, ecological production processes, and the design, manufacture and commercialization of environmentally sustainable products requires that CEOs have a greater degree of power in order to support projects that do not entail a higher return in the short and medium terms. Furthermore, our findings also indicate that the negative economic effect of eco-innovation reverses in the fourth and fifth years after environmental innovations were promoted. These results are robust for different specifications of firm performance, considering this magnitude both from the point of view of the stock market and the returns on assets and shareholders.

Our study contributes to prior literature in several ways. Firstly, we contribute to the CSR literature by providing empirical evidence of the role that individual-level factors play in shaping corporate environmental proactive strategies. Thus, our findings confirm the arguments regarding the influence of the CEO's individual characteristics on such strategies (Arena et al. 2018; García-Sánchez et al. 2020c) and, specifically, we show that CEO power is positively related to eco-innovation. Although previous research has analyzed the effect of several CEOs' characteristics and attributes (i.e., religious beliefs, hubris, compensation) on environmental innovation, to the authors' knowledge, no study has explored the impact of CEO power. In this sense, our findings contribute to the debate about whether the concentration of power in the figure of the CEO may be harmful or beneficial to the firm by showing that CEO power has a positive impact on the firm's long-term value promoting environmental innovative projects.

We also contribute to CSR literature by analyzing the relationship between environmental innovation and financial performance, another controversial topic on which no conclusive findings have been obtained in previous studies. Our results indicate that, although eco-innovation projects have a negative effect on financial performance in the short term, leading to a lesser financial performance during the three years following their implementation, afterward their impact on financial performance changes and it increases in the fourth and the fifth years after proactive environmental strategies were implemented. Accordingly, our findings suggest that eco-innovations have a positive influence on financial performance in the long term.

Secondly, we contribute to the literature on eco-innovation examining these relationships for three different types of environmental innovations (i.e., innovations related to the use of clean technologies, innovations related to ecological production processes, and innovations related to the design, manufacture and commercialization of environmentally sustainable products). Thus, we extend and reinforce prior findings regarding the impact of eco-innovation on business performance. Finally, from a theoretical viewpoint, our research contributes to the upper echelon theory by providing evidence on the role that an attribute of the CEO (i.e., power) plays in relation to a specific type of strategic decisions (i.e., those concerning investments in environmental innovation), which has not been analyzed in prior studies. Thus, our findings add to the upper echelon research empirical evidence regarding a new attribute that characterizes the profile of the CEO of environmentally proactive firms. Furthermore, we show that, although from the agency theory perspective, greater CEO power favors managerial entrenchment, which could inhibit those investments that have a long-term horizon and involve high risks, such as long-term capital investments in environmental projects, leading to a decline in environmental performance (De Villiers et al. 2011; Harper and Sun 2019; Sheikh 2019), the greater managerial ability commonly associated with CEO power (Finkelstein 1992; Han et al. 2016) counteracts this tendency, since more able CEOs are confident about their capability to compensate for the potential negative results from such investments (García-Sánchez and Martínez-Ferrero 2019). In this sense, our results confirm prior findings regarding a "bright side" of CEO power with regard to corporate sustainability (Walls and Berrone 2017; Li et al. 2018; Velte 2019). Finally, we adopt an international approach, using data from 70 countries and 10 activity sectors, and consider a long period of analysis (16 years), which makes our results more robust.

This paper's findings have some practical implications for companies, stakeholders and policy makers. For companies, our findings indicate that, to the extent that CEOs have a high environmental awareness, empowering the CEO represents an opportunity rather than a threat to increase firm value, and, more importantly, our findings suggest that firms should avoid a "myopic focus" and take a long-term perspective, promoting proactive environmental strategies, such as eco-innovation, if they really want to improve their competitiveness. Similarly, from the investors' perspective, the findings provide confidence that eco-innovation projects will have a positive impact on firm value. Finally, the results offer some guidelines that could assist policy makers to establish incentives for developing eco-innovation, for example, stimulating the accumulation of power in the figure of the CEO or establishing policies aimed at raising managerial awareness regarding the environmental and economic benefits that eco-innovation projects can generate.

This paper is structured as follows: after this introduction, the development of the research hypotheses on the relationship between environmental innovation and (i) CEO power and (ii) financial performance is presented in the following section. The Section 3 describes the empirical framework of the study (sample, variables, models, and analysis techniques), whereas the Section 4 presents and discusses the main findings as well as some robustness analysis. Finally, the last section summarizes the conclusions and main implications of the study.

## 2. Theoretical Background and Research Hypotheses

### 2.1. Managerial Concern as a Driver of Eco-Innovarion

Among the different definitions of eco-innovation proposed in literature, this study adopts the following: eco-innovations are "innovations that consist of new or modify processes, practices, systems and products which benefit the environment and so contribute to environmental sustainability" (Oltra and Saint-Jean 2009, p. 1). Although eco-innovations share many characteristics with general innovation, as both involve risks and temporal trade-offs (Brossard et al. 2013; Del Río et al. 2016), several authors stress that eco-innovations are usually riskier, more complex, and more uncertain than general innovations and their capital costs are higher (Berrone et al. 2013; Arena et al. 2018; Cecere et al. 2020). Furthermore, eco-innovations have as a distinctive feature the "double externality" problem (Rennings 2000), which reduces the firm's incentives to develop this type of innovations (Cai and Li 2018). Nevertheless, eco-innovations also generate several economic benefits for firms allowing them to achieve competitive advantages (Porter and Linde 1995; Demirel and Kesidou 2011) and enhance their image/reputation (Bigliardi et al. 2012).

Overall, the drivers of eco-innovation encompass both the firm's resources and capabilities that determine how it can compete and the external pressures that it faces (García-Sánchez et al. 2020b). Among the first group of drivers, managerial environmental concern has been found a major driving force of eco-innovation (Qi et al. 2010; Agan et al. 2013; Hojnik and Ruzzier 2016). In this sense, to the extent that managers are more environmentally conscious and concerned about the negative impacts on the natural environment generated by their firms' operations, they will tend to promote eco-innovation projects. Furthermore, managers concerned about their reputation might also be more inclined to support this type of proactive environmental strategy, either to avoid the damages derived from their firms' environmental misconduct and compliance problems (Berrone et al. 2013) or to convey a better image.

### 2.2. Environmental Innovation and Chief Executive Officer (CEO) Power

CEO power has been the subject of considerable debate in the literature, existing two opposite views according to which the accumulation of power in the figure of the CEO can be harmful or beneficial for the firm (Sah and Stiglitz 1986). On the one hand, following the agency theory's tenets (Jensen and Meckling 1976), higher CEO power increases managerial entrenchment (DeAngelo and Rice 1983) and, therefore, the likelihood of CEOs pursuing their own benefit at the expense of shareholders' interest (Sheikh 2018). Furthermore, powerful CEOs tend to be overconfident and disregard expert advice and thus make unilateral decisions that may imply costly mistakes (Eisenhardt and Bourgeois 1988; Haleblian and Finkelstein 1993) with a negative impact on firm value (Bebchuk et al. 2011; Landier et al. 2013). On the other hand, powerful CEOs are more able to answer to changes in the environment, making quick and timely decisions aimed at protecting or enhancing firm value (Finkelstein and D'aveni 1994; Boyd 1995).

As regards the relationship between CEO power and firm innovation, again two different arguments can be set out. The first one posits a negative relationship between CEO power and eco-innovation, so that higher CEO power results in less innovation. Innovation projects tend to be highly risky and uncertain and require to invest greater financial resources in the short run to generate (uncertain) returns in the long run (David et al. 2001), which may produce aversion to them in CEOs. Accordingly, from the agency theory perspective, to the extent that the CEO enjoys a higher degree of power in the firm, managerial entrenchment increases and, consequently, she/he will use her/his power to inhibit the development of innovations.

On the contrary, the second view posits a positive association between CEO power and innovation, so that higher CEO power favors firm innovation. Two reasons support this view. In the first place, as noted by Sheikh (2018, p. 37), "CEO power is a multidimensional concept", by which exist several sources of power: structural, ownership, expert,

and prestige (Finkelstein 1992). In this sense, the CEO's managerial ability (expert dimension of CEO power) is associated with more positive career prospects and professional opportunities (García-Sánchez and Martínez-Ferrero 2019; García-Sánchez et al. 2020c), which reduces her/his risk aversion. In the second place, more capable CEOs are also more self-assured about their performance in complex situations (Griffin and Tversky 1992), which makes them more inclined to promote innovation projects.

A few studies have analyzed the relationship between CEO power and firm innovation by showing a positive association between them. Thus, Sariol and Abebe (2017) reported a positive effect of CEO power on organizational innovation; whereas Galasso and Simcoe (2011) and Hirshleifer et al. (2012) found a positive impact of CEO overconfidence on innovation. Chen (2014) reported a moderating role of CEO power in the relationship between board and innovation, in such a way that the board of directors is more likely to support innovation projects in the presence of powerful CEOs. Finally, Sheikh (2018) demonstrated that CEO power is positively associated with firm innovation, but only in markets characterized by high competition.

With regard to eco-innovation, we consider that, to the extent that the CEO is environmentally conscious and assumes her/his company's social responsibility, she/he will encourage this type of innovations in spite of the negative effect that these investments may have on business financial performance in the short term. Thus, given that powerful CEOs are less constrained by board monitoring and have the necessary autonomy to make decisions and impose their own criterion, it could be expected that a higher degree of CEO power favors eco-innovation.

Accordingly, we expect a positive effect of CEO power on eco-innovation and, consequently, and propose the following hypothesis:

**Hypothesis 1 (H1).** *There is a positive relationship between CEO power and environmental innovation.*

*2.3. Environmental Innovation and Financial Performance*

Eco-innovations not only entail environmental benefits, but also are a source of competitiveness for firms (Porter and Linde 1995; Demirel and Kesidou 2011; Bigliardi et al. 2012; García-Sánchez et al. 2020a). Indeed, several studies have showed that successful environmental innovations allow companies to diminish costs and risks (Hart and Ahuja 1996; Ambec and Lanoie 2008), increase sales and market share both in current markets and in new markets (Kim et al. 2019; Oh et al. 2020), improve stakeholder relationships and firm reputation (Brammer and Pavelin 2006; Cheng and Shiu 2012), and enhance productivity and margins (Sammer and Wüstenhagen 2006; Ruben et al. 2009; Ruben and Fort 2012), among other competitive benefits. To the extent that the overall benefits exceed the costs and risks involved, eco-innovations will have a positive impact on financial performance (Long et al. 2017). Accordingly, the achievement of a higher financial performance could be a motivation behind environmental innovation (Li et al. 2017).

The study of the effect of eco-innovation on financial performance has received considerable attention in the literature, although empirical results are not conclusive existing two opposite trends (Przychodzen and Przychodzen 2015; Alos-Simo et al. 2020; Oh et al. 2020). The majority of empirical evidence documents a positive association between eco-innovation and financial performance (Hojnik et al. 2018). These studies have stressed the potential economic benefits for companies associated with eco-innovation. For example, some authors showed that customers prefer eco-labeled products (Martinez-del-Rio et al. 2015) and are more willing to pay for them (Sammer and Wüstenhagen 2006), so that eco-labeling results in an increase of market share and sales (Oh et al. 2020). Cost savings (Ambec and Lanoie 2008; Kanda et al. 2018) as well as the reduction in the risks and costs of penalties and litigation derived from eco-innovation (Hart and Ahuja 1996) also contribute to enhance firm profitability (Ma et al. 2018). Furthermore, environmental innovation brings reputational benefits (Semenova and Hassel 2008; Berrone et al. 2013) and

is positively valued by investors who acknowledge the positive impact of eco-innovation on the firm's long-term value (García-Sánchez et al. 2020a, 2020b). In this sense, Cheng et al. (2014) report a positive association between different types of eco-innovation and several measures of business performance (i.e., return on investment (ROI), market share, profitability, and sales) in Taiwanese firms and, likewise, in the Hungarian context, Przychodzen and Przychodzen (2015) showed that eco-innovation leads to higher return on assets (ROA) and return on equity (ROE).

Conversely, other studies reveal that eco-innovation not always has a positive impact on business performance because its potential benefits could be annulled (McWilliams and Siegel 2001; Martinez-del-Rio et al. 2015). In this sense, some authors (Boons and Wagner 2009; Carrillo-Hermosilla et al. 2009) pose that successful eco-innovation requires high technical skills and technological competencies and involves a strong initial investment, which can decrease profitability. Bigliardi et al. (2012) found that firms engaged with eco-innovation have a high degree of turnover per worker, whereas Fernando et al. (2010) observed that these proactive strategies do not lead to an increase in the firm's market valuation, and Rexhäuser and Rammer (2014) found that not all eco-innovations bring about positive economic returns for firms. More recently, García-Sánchez et al. (2020b) showed that, although environmental innovation strategies are well valued by capital markets, they do not entail higher returns leading to lower profitability.

As Alos-Simo et al. (2020) noted, eco-innovation projects take time to show tangible, observable results, which could explain that, although they have an immediate negative effect on performance, some years after their implementation they do contribute to improve performance. In this sense, some authors (Clemens 2006; Bigliardi et al. 2012) observed that eco-innovations have a negative impact on performance in the first year but a positive effect after two years. In a similar way, in the case of Spanish metal companies, Amores-Salvadó et al. (2014) found a positive effect of environmental innovations on the firm's operating profit with 2 years lag, and Ma et al. (2018) also documented a positive effect of eco-innovation on Chinese firms' economic performance after 2 years lag.

Considering that the first trend has been supported by most prior empirical findings (Hojnik et al. 2018), we expect that, even if in the short term the impact of eco-innovation on financial performance may be negative, in the medium and long term it has positive effects on financial performance and, consequently, we formulate the following hypothesis:

**Hypothesis 2 (H2).** *The positive relationship between environmental innovation and financial performance occurs in the medium and long terms.*

### 3. Methodology

#### 3.1. Sample

The data used in this research was obtained from Thomson Reuters database. The procedure used to determine the sample and the analysis period consisted in the selection of all listed companies worldwide with available information for the environmental innovation variables used in the empirical models to be estimated. Subsequently, those companies that did not have information about the remaining variables considered in the models were eliminated. Finally, the companies with a frequency under eight years were eliminated, given that this value was necessary for determining future returns associated with environmental innovations and controlling endogeneity problems.

The sample consisted of 4863 companies over a period of 16 years (2002–2017), which corresponded to an unbalanced data panel of 42,813 observations. The sample was unbalanced due to the fact that the information of all the companies was not available in all years. The sample companies were located in 70 different countries and belonged to 10 activity sectors.

### 3.2. Models and Variables

The first research hypothesis aims to examine the impact of CEO power on environmental innovation. The following model [Equation (1)] is proposed where "Env_Inno" is returned in relation to the power of the CEO and several control variables that represent the main firm characteristics, the effectiveness of the board of directors, and other characteristics of the institutional environment.

$$\text{Env\_Inno}_{i,t} = \beta_0 + \beta_1\text{CEOPower}_{i,t} + \sum_{j=2}^{16} \beta_j\text{Control}_{i,t} + \beta_{17}\text{Industry}_{i,t} + \beta_{18}\text{Country}_{i,t} + \beta_{19}\text{Year}_t + \mu_{it} + \eta_i \quad (1)$$

In order to determine the effect of eco-innovation on financial performance, we estimate the Equation (2), where performance is represented by Tobin's Q (QTobin), which is computed though the relationship between the market value of the company and the replacement value of its total assets. The model also includes a retard of the endogenous variable. Likewise, we will include several control variables to avoid biased results.

$$\text{QTobin}_{i,t} = \varphi_0 \; + \varphi_1\text{QTobin}_{i,t-1} + \varphi_2\text{Env\_Inno}_{i,t} + \sum_{j=3}^{17} \varphi_j\text{Control}_{i,t} + \varphi_{18}\text{Industry}_{i,t} + \varphi_{19}\text{Country}_i + \varphi_{20}\text{Year}_t$$
$$+ \varphi_{21}\text{Crisis}_t + \mu_{it} + \eta_i \quad (2)$$

Each company is identified by i, and t refers to the year. β and φ are the parameters to be estimated.

In relation to Equation (1), the "Env_Inno" variable corresponds to an ordinal variable that takes values between 0 and 3 to identify whether the company has implemented environmental innovation strategies related to: (i) the use of clean technologies in their production processes, (ii) an ecological process in the design and manufacture of products, and (iii) the design, production and commercialization environmentally responsible products. This variable reflects the overall level of environmental innovation of a company associated with the different types of eco-innovation. Thus, in the case that the company carries out only one type of eco-innovation, this variable would take the value 1; if the company develops two types of eco-innovations, the variable would take the value 2; and, in the case of a company involved in the three types of eco-innovation, the variable would take the maximum value of 3.

Prior literature (Finkelstein 1992; Bebchuk et al. 2011; Chen 2014; Han et al. 2016; Li et al. 2017; Arena et al. 2018; Sheikh 2018) has showed that CEO power can come from several sources (i.e., structural, ownership, expert, and prestige) and, consequently, can be measured by using different proxies (e.g., CEO-chair duality, CEO share ownership, CEO pay slice, CEO tenure, CEO connectedness, CEO founder status). In this paper, following Garcia-Sanchez et al. (2020d), we treat CEO power as an index that reflects three dimensions related to CEO structural power. Thus, CEO power is represented by an indicator that takes values between 0 and 3 points, assigning one point to each of the following three situations: (i) the CEO is a member of the board of directors, (ii) she/he exercises the functions of chairman of the board, and (iii) the percentage of executive directors on the board is higher than the average. The final score is the sum of the scores obtained in these three conditions.

Regarding the control variables, following previous studies (García-Sánchez et al. 2020a; García-Sánchez et al. 2020b; García-Sánchez et al. 2021), we have included a broad set of control variables representing the firm's capabilities and resources, monitoring mechanisms, and institutional pressures. The inclusion of these variables allows us to avoid biased results because of the impact they have on the development of proactive environmental strategies by companies.

According to the resource-based theory (Keshminder and Río 2019), larger and more profitable firms have a greater volume of resources that can be devoted to eco-innovation projects (Bigliardi et al. 2012; Przychodzen and Przychodzen 2015). Likewise, firm age also favors eco-innovation, as older companies have accumulated knowledge and experience related to the development of other environmental strategies (Rehfeld et al. 2007) and

often have greater access to external financing (Johnson and Lybecker 2012). Furthermore, investments in physical capital and R&D benefit the development of the technological capabilities that are necessary to carry out eco-innovation projects (Horbach 2008; Cainelli et al. 2015). Finally, internationalization stimulates companies to eco-innovate as the learning processes related to internationalization activities favor innovation and, moreover, companies have to meet foreign environmental regulations (Hojnik et al. 2018). All these factors are represented in our models through the following control variables: Size (logarithm of assets); ROA (economic profitability represented by the return on assets ratio); Leverage (proportion of debt compared to the company's own resources); Interna (the level of internationalization of operations represented by the percentage of sales in international markets); capital expenditure (CAPEX, intensity of investment in physical capital with respect to total sales); research and development (R&D, intensity of investment in R&D&i with respect to total sales); WC (the working capital or liquidity of the firm); and F_Age (the age of the company).

Given that the approval of corporate strategies and the monitoring of the management team are among the functions of the board of directors (García-Sánchez et al. 2021), several variables representing board attributes were also included in the models. Thus, considering the key role played by independent directors in monitoring management (Fama and Jensen 1983), their presence on the board will affect the extent to which managerial entrenchment is limited. Furthermore, the monitoring role and independence of the board of directors are also affected by its size. With regard to the influence of board independence on eco-innovation, García-Sánchez et al. (2021) found that it has a positive effect on eco-innovation and eco-design strategies. According to the upper echelon theory, the decision-makers' characteristics, independently they are managers or directors, influence business strategies (Hambrick and Mason 1984). In this regard, literature documents the effect of gender differences on personality (Van der Walt and Ingley 2010), expectations, risk-taking behavior (Srinidhi et al. 2011), and sensitivity toward ethical, environmental and social concerns (Labelle et al. 2010; Carli and Eagly 2016). Specifically, some studies (Liao et al. 2019; Nadeem et al. 2020) found that female directors are more inclined to promote environmental innovation. Finally, CSR committees have the function of improving the firm's environmental and social behavior (García-Sánchez et al. 2019), affecting positively eco-innovation (Liao et al. 2015). Accordingly, four control variables were included in the models representing the size of the board of directors (Bsize), measured by the total number of directors; the level of independence of the board (Bindep), measured by the percentage of external directors on the board; board diversity (Bwomen), measured as the percentage of women directors on the board; and the existence of a CSR committee (CSRCommittee), measured as a dummy variable that takes a value of 1 if the company has created a CSR committee and 0 otherwise.

Institutional pressures are identified with the regulatory requirements regarding environmental protection at country (ERRI) and sector (IENVPI) levels, as well as the efficiency of the judicial system in relation to compliance with such pressures (EJ) proposed by Porta et al. (1998). ERRI corresponds to the ranking of national environmental regulations proposed by Esty and Porter (2002), whereas IENVPI corresponds to the sectoral ranking in terms of institutional pressures proposed by Amor-Esteban et al. (2018, 2019b).

Regarding Equation (2), QTobin variable corresponds to the quotient between the market value of a company's assets and their replacement cost. Its value is a good predictor of the company's economic conditions since, if it is less than 1, the value of the assets would be lower than its replacement cost and, therefore, new investments in similar assets would not be profitable. If, on the other hand, it is greater than 1, it would be a signal for a greater investment in similar assets. Thus, this variable indicates the value assigned to corporate proactive strategies by the capital market agents (Surroca et al. 2010; García-Sánchez et al. 2020b).

As the independent variable in Equation (2), we consider Env_Inno. In this model, in addition to the variables related to firm characteristics, following García-Sánchez et al. (2020b, 2021), we include sectoral munificence level (Munif), dividend per share (Div) and two scores that determine the company's corporate governance practices (CGScore) and social commitment (SocialScore). Munificent industries are characterized by benefiting from greater tax incentives and better financing plans for investments in initiatives such as environmental innovation projects, so that managers of firms belonging to these industries have more incentives and resources to implement eco-innovation strategies (García-Sánchez et al. 2020b). We also control the level of dividend that companies have distributed as it reflects the firm's policy of profit distribution. Finally, institutional pressures are reflected in the National Corporate Social Responsibility Practices Index (NCSRPI) composite indicator proposed by Amor-Esteban et al. (2019a).

Additionally, we include ordinal variables to control the effect of the Industry, Country and Year due to we use an unbalanced panel data in which the information of all companies is not available in all years. Furthermore, prior studies have showed that both the sector's and the country's technological trajectories affect eco-innovation (Cai and Zhou 2014; Hojnik et al. 2018; Demirel and Kesidou 2019). Finally, following Gallego-Álvarez et al. (2014), we take into consideration the possible incidence of periods of economic recession that could affect the firm's capability to eco-innovate. In this regard, although the financial crisis did not affect symmetrically all countries, industries, and even companies and its impact on innovation remains unclear, as the economic downturn can boost or inhibit innovation (Hausman and Johnston 2014; Zouaghi et al. 2018), we decided to include an ordinal variable representing unfavorable economic periods (Crisis) in the models to reflect how dynamic capabilities (defined by Teece et al. (1997, p. 519) such as "the ability to integrate, build and reconfigure internal and external competencies to address rapidly changing environments") affect the CEO's attitude towards eco-innovation.

*3.3. Analysis Techniques*

The nature of the dependent variables demands the employment of different econometric methodologies for panel data. Thus, Equation (1) will be estimated by using an ordinal regression, whereas, following García-Sánchez et al. (2020b), Equation (2) will be estimated by using the dynamic estimator in two-stages based on the generalized method of moments (GMM) proposed by Arellano and Bond (1991), available in Stata through Roodman (2009).

The independent and control variables are introduced with a lag in order to correct for causality and endogeneity problems. Furthermore, following García-Sánchez et al. (2020a, 2020b, 2021) both models also incorporate an error term divided into two elements: η, which refers to the specific effect of the company and allows us to control the unobservable heterogeneity, and μ, which refers to the traditional random perturbance.

**4. Results**

*4.1. Descriptive Analysis*

Table 1 shows the descriptive statistics for the variables considered in the analysis. In this regard, it can be observed that the level of environmental innovation stands at 0.504, with a standard deviation of 0.842. This value indicates that, on average, the sample companies have implemented one of the three environmental innovation strategies considered in this study (i.e., the use of clean technologies in their production processes, an ecological process in the design and manufacture of products, and the design, production and commercialization environmentally responsible products). More specifically, almost 70% of the companies have not invested in any environmental innovation project, 14.4% of the companies have implemented only one of these strategies, 13.1% of the companies have implemented two types of environmental innovation strategies, and only 3.3% of the companies have promoted global environmental innovation.

Tobin's Q reaches an average value higher than 1, which means that the market value of the firms' assets is higher than their replacement cost. This may be indicative of a positive valuation of these investments by the market capital agents.

The average CEO power is close to 2, which indicates a medium–high degree of power concentrated on the CEO.

**Table 1.** Descriptive statistics.

| Variable | Mean | Std Dev |
|---|---|---|
| Env_Inno | 0.504 | 0.842 |
| CEOPower | 1.915 | 0.737 |
| QTobin | 1.751 | 112.614 |
| Size | 15.738 | 3.018 |
| ROA | 4.274 | 16.159 |
| Leverage | 0.845 | 0.889 |
| Interna | 0.307 | 0.682 |
| CAPEX | 0.112 | 0.510 |
| R&D | 0.159 | 0.694 |
| WC | 4.550 | 1.530 |
| F_Age | 33.111 | 30.692 |
| Bsize | 10.242 | 3.655 |
| Bindep | 0.514 | 0.302 |
| Bwomen | 0.114 | 0.115 |
| ERRI | 0.920 | 0.642 |
| IENVPI | −0.014 | 0.902 |
| EJ | 9.217 | 1.543 |
| Munif | −0.224 | 0.428 |
| Div | 45.767 | 612.356 |
| CGScore | 51.343 | 30.910 |
| SocialScore | 14.302 | 18.536 |
| NCSRPI | −1.820 | 8.950 |

| Variable | Frecuencies |
|---|---|
| CSRCommittee | 0.425 |
| Env_Innov | |
| 0 | 0.692 |
| 1 | 0.144 |
| 2 | 0.131 |
| 3 | 0.033 |

Table 2 shows the Pearson coefficients. As can be seen, the coefficients are not high, which suggests that there are no multicollinearity problems.

**Table 2.** Bivariate correlations (*** $p < 0.01$, ** $p < 0.05$, * $p < 0.1$).

| | | 1 | 2 | 3 | 4 | 5 | 6 | 7 | 8 | 9 | 10 | 11 | 12 | 13 | 14 | 15 | 16 | 17 | 18 | 19 | 20 | 21 | 22 | 23 |
|---|---|---|---|---|---|---|---|---|---|---|---|---|---|---|---|---|---|---|---|---|---|---|---|---|
| 1 | Env_Inno | 1 | | | | | | | | | | | | | | | | | | | | | | |
| 2 | CEOPower | 0.00 | 1 | | | | | | | | | | | | | | | | | | | | | |
| 3 | QTobin | −0.09 *** | 0.00 | 1 | | | | | | | | | | | | | | | | | | | | |
| 4 | Size | 0.26 *** | 0.08 *** | −0.02 *** | 1 | | | | | | | | | | | | | | | | | | | |
| 5 | ROA | −0.01 | 0.00 | −0.06 *** | 0.03 *** | 1 | | | | | | | | | | | | | | | | | | |
| 6 | Leverage | 0.00 | −0.01 | 0.00 | 0.00 | 0.00 | 1 | | | | | | | | | | | | | | | | | |
| 7 | Interna | 0.07 *** | 0.00 | 0.00 | 0.02 *** | 0.01 *** | 0.00 | 1 | | | | | | | | | | | | | | | | |
| 8 | CAPEX | −0.01 ** | 0.00 | 0.00 | −0.02 *** | −0.03 *** | −0.01 | 0.00 | 1 | | | | | | | | | | | | | | | |
| 9 | R&D | −0.01 ** | −0.01 | 0.02 *** | −0.02 *** | −0.05 *** | −0.01 | −0.03 *** | 0.02 *** | | | | | | | | | | | | | | | |
| 10 | WC | 0.04 *** | 0.02 *** | 0.00 | 0.08 *** | 0.00 | 0.00 | 0.00 | 0.00 | 0.00 | 1 | | | | | | | | | | | | | |
| 11 | F_Age | 0.20 *** | −0.02 *** | −0.01 | 0.22 *** | 0.01 *** | 0.00 | 0.08 *** | −0.01 *** | −0.02 ** | 0.01 *** | 1 | | | | | | | | | | | | |
| 12 | Bsize | 0.06 *** | 0.06 *** | −0.02 *** | 0.16 *** | 0.01 | 0.00 | 0.00 | 0.00 | 0.00 | 0.00 | 0.07 *** | 1 | | | | | | | | | | | |
| 13 | Bindep | −0.05 *** | −0.61 *** | 0.00 | −0.18 *** | −0.01 | 0.01 | −0.01 | −0.02 *** | 0.01 | −0.02 *** | 0.00 | −0.13 *** | 1 | | | | | | | | | | |
| 14 | Bwomen | 0.05 *** | −0.17 *** | 0.01 | −0.09 *** | 0.00 | 0.00 | −0.01 ** | −0.02 *** | 0.01 * | −0.02 *** | 0.04 *** | 0.05 *** | 0.27 *** | 1 | | | | | | | | | |
| 15 | CSRCommittee | 0.21 *** | 0.04 *** | 0.00 | 0.10 *** | 0.00 | −0.01 | 0.03 *** | 0.01 | −0.01 | 0.01 | 0.07 *** | 0.16 *** | −0.01 | 0.15 *** | 1 | | | | | | | | |
| 16 | ERRI | 0.01 | −0.12 *** | 0.00 | −0.39 *** | −0.01 *** | 0.00 | 0.07 *** | −0.06 *** | 0.01 * | −0.06 *** | 0.03 *** | −0.05 *** | 0.13 *** | 0.09 *** | −0.02 *** | 1 | | | | | | | |
| 17 | IENVPI | 0.18 *** | 0.01 | 0.00 | −0.03 *** | 0.00 | 0.00 | 0.11 *** | 0.02 *** | −0.01 | 0.02 *** | 0.10 *** | −0.02 *** | −0.02 *** | −0.05 *** | 0.06 *** | −0.07 *** | 1 | | | | | | |
| 18 | EJ | −0.06 *** | −0.09 *** | 0.00 | −0.37 *** | −0.02 *** | 0.00 | 0.01 ** | −0.09 *** | 0.00 | −0.09 *** | −0.04 *** | −0.08 *** | 0.12 *** | 0.05 *** | −0.05 *** | 0.78 ** | −0.06 *** | 1 | | | | | |
| 19 | Munif | 0.21 *** | 0.03 *** | 0.00 | 0.21 *** | 0.03 *** | 0.01 | 0.00 | 0.03 *** | −0.02 *** | 0.03 *** | 0.13 *** | −0.01 | −0.05 *** | 0.04 *** | 0.10 *** | −0.06 *** | 0.15 *** | −0.06 *** | 1 | | | | |
| 20 | Div | 0.01 *** | 0.01 *** | 0.00 | 0.14 *** | 0.00 | 0.00 | −0.01 ** | 0.12 *** | 0.00 | 0.12 *** | −0.01 * | −0.02 *** | −0.03 *** | −0.04 *** | 0.00 | −0.07 *** | 0.00 | −0.08 *** | 0.04 *** | 1 | | | |
| 21 | CGScore | −0.04 *** | −0.33 *** | 0.01 | −0.21 *** | 0.00 | 0.00 | −0.01 | −0.04 *** | 0.01 | −0.04 *** | −0.01 | −0.02 *** | 0.64 *** | 0.30 *** | 0.09 *** | 0.11 *** | −0.02 *** | 0.12 *** | −0.04 *** | −0.07 *** | 1 | | |
| 22 | SocialScore | −0.02 *** | 0.01 | 0.00 | −0.02 *** | 0.00 | −0.01 | 0.01 ** | 0.01 | −0.01 | 0.01 | −0.05 *** | 0.00 | 0.00 | 0.00 | 0.00 | −0.02 *** | −0.01 | −0.02 *** | −0.01 ** | 0.00 | 0.01 | 1 | |
| 23 | NCSRPI | 0.04 *** | 0.13 *** | 0.00 | −0.01 *** | 0.01 *** | 0.01 * | 0.15 *** | −0.02 *** | −0.01 ** | −0.02 *** | 0.09 *** | −0.04 *** | −0.10 *** | 0.00 | 0.05 *** | 0.21 *** | 0.06 *** | −0.02 *** | 0.05 *** | 0.02 *** | −0.07 *** | 0.01 * | 1 |

*4.2. Main Results*

Table 3 shows the estimates for Equations (1) and (2). It can be seen that the CEOPower variable has a significant impact (coeff. = 0.0290) on the level of corporate environmental innovation for a confidence level of 95%. This result allows us to confirm Hypothesis 1 which posited that the accumulation of power in the figure of the CEO can be a determinant of the company's decision to eco-innovate and, specifically, that powerful CEOs favor eco-innovation. This result is in line with those obtained by Sheikh (2018) with regard to general innovation. In this sense, although prior studies pose that CEO power causes agency problem, reduces the effectiveness of the board's monitoring role, and reduces the likelihood of approving long-term capital investments in environmental projects, leading to a decline in environmental performance (De Villiers et al. 2011; Harper and Sun 2019; Sheikh 2019), our findings are in line with other authors who demonstrate the opposite (Walls and Berrone 2017; Li et al. 2018; Velte 2019). Indeed, this finding suggests that conferring more power to the CEO may have a positive impact on the development of eco-innovations, because she/he has the necessary autonomy to make decisions about investing in eco-innovation projects, regardless of the effect that these investments may have on business financial performance.

The results obtained for Equation (2) allow us to partially accept Hypothesis 2 which posited a positive association between environmental innovation and financial performance in the medium and long term. As can be seen in Table 3, environmental innovations have a negative impact on Tobin's Q in the year in which they are carried out (coeff. = −0.0959). This negative effect is maintained during the three years that followed the implementation of the eco-innovation strategies (t + 1: coeff. = −0.0274; t + 2: coeff. = −0.00914; t + 3: coeff. = −0.0272). However, from the fourth year that followed the implementation of the eco-innovation strategies, the impact of eco-innovation on Tobin's Q is positive (t + 4: coeff. = 0.0727; t + 5; coeff. = −0.223). This finding is in line with those obtained by Ramanathan et al. (2010) and García-Sánchez et al. (2020b) regarding the negative impact of eco-innovation on firm-performance in the short run and confirms the arguments regarding the existence of a lag between the implementation of an eco-innovation strategy and its tangible results, which would explain the fact that the potentially positive effects of eco-innovation on performance come into fruition at a later date (Alos-Simo et al. 2020). However, although prior research reports that eco-innovation positively affects operating profit (Amores-Salvadó et al. 2014) and economic profit (Ma et al. 2018) with two years lag, our findings indicate that a longer period (four years) is necessary to obtain a positive effect of eco-innovation on financial performance, perhaps as a consequence that it takes time to recover the initial costs associated with the complexity of innovative environmental projects and to achieve that stakeholders accept these strategies. Therefore, our findings indicate that environmental innovation strategies only generate value in the long term and, therefore, the Hypothesis 2 that posited that eco-innovations have positive effects on financial performance in the medium and long term only is valid in the case of a long-term horizon.

Additionally, Table 3 shows that innovative environmental investments are promoted to a greater extent by the largest and oldest companies and those with the greatest presence in international markets. However, it should be noticed that the sign of these and other control variables changes when the lag in Tobin's Q changes, which requires further analysis. Furthermore, eco-innovation investments are also strengthened by regulatory pressure at national and sectoral level as well as by the existence of a CSR committee on the board of directors. In this case, our results contrast with those obtained by García-Sánchez et al. (2021), who found a positive but not statistically significant effect of the existence of a CSR committee on eco-innovation.

**Table 3.** Main results (*** $p < 0.01$, ** $p < 0.05$, * $p < 0.1$).

| | | QTobin | | | | | |
|---|---|---|---|---|---|---|---|
| | Env_Inno | t | t + 1 | t + 2 | t + 3 | t + 4 | t + 5 |
| | Coeff. (Std. Err.) | Coeff. (Std. Err.) | Coeff. (Std. Err.) | Coeff. (Std. Err.) | Coeff. (Std. Err.) | Coeff. (Std. Err.) | Coeff. (Std. Err.) |
| QTobin$_{t-1}$ | | 0.436 *** (0.00304) | 1.072 *** (0.00619) | 0.250 *** (0.00169) | −0.0249 *** (0.000996) | −0.607*** (0.00142) | −0.637*** (0.000735) |
| CEOPower | 0.0290 ** (0.0136) | | | | | | |
| Env_Inno | | −0.0959 *** (0.00600) | −0.0274 *** (0.00681) | −0.00914 (0.00659) | −0.0272 *** (0.00623) | 0.0727 *** (0.00351) | 0.223 *** (0.0102) |
| Size | 0.115 *** (0.00745) | −1.053 *** (0.0176) | 0.415 *** (0.0224) | −0.386 *** (0.0133) | −0.567 *** (0.00981) | −0.194 *** (0.0115) | 0.121 *** (0.0181) |
| ROA | −0.000282 (0.000695) | 0.0125 *** (0.000377) | 0.00261 *** (0.000410) | −0.00785 *** (0.000545) | 0.0714 *** (0.000819) | 0.0125 *** (0.000381) | −0.0177 *** (0.000403) |
| Leverage | $1.11 \times 10^{-6}$ $(3.61 \times 10^{-6})$ | $1.11 \times 10^{-6}$ *** $(2.11 \times 10^{-7})$ | $1.58 \times 10^{-6}$ *** $(1.08 \times 10^{-7})$ | $5.48 \times 10^{-6}$ *** $(2.86 \times 10^{-7})$ | $-6.89 \times 10^{-8}$ $(3.00 \times 10^{-7})$ | $5.91 \times 10^{-6}$ *** $(2.00 \times 10^{-6})$ | $-1.27 \times 10^{-5}$ *** $(1.55 \times 10^{-6})$ |
| Inter | 0.00394 *** (0.000389) | 0.00350 *** (0.000260) | −0.00677 *** (0.000347) | 0.00456 *** (0.000260) | −0.00905 *** (0.000280) | −0.00633 *** (0.000233) | 0.00296 *** (0.000281) |
| CAPEX | $1.19 \times 10^{-6}$ $(1.13 \times 10^{-6})$ | 0.00483 *** $(4.81 \times 10^{-5})$ | −0.00733 *** $(8.38 \times 10^{-5})$ | 0.000571 *** $(2.60 \times 10^{-5})$ | −0.00757 *** (0.000392) | −0.0180 *** (0.000148) | −0.0153 *** (0.000284) |
| R&D | $-1.00 \times 10^{-6}$ * $(5.59 \times 10^{-7})$ | $-4.04 \times 10^{-5}$ *** $(1.63 \times 10^{-6})$ | 0.000104 *** $(1.51 \times 10^{-5})$ | $-6.95 \times 10^{-5}$ *** $(1.25 \times 10^{-5})$ | 0.00692 *** (0.000122) | 0.00363 *** $(3.16 \times 10^{-5})$ | −0.000620 *** $(6.77 \times 10^{-5})$ |
| WC | 0.000 (0.000) | −0.001 *** (0.000) | 0.001 *** (0.000) | 0.001 *** (0.000) | −0.001 *** (0.000) | $-2.92 \times 10^{-10}$ *** (0.000) | $-4.60 \times 10^{-10}$ *** (0.000) |
| F_Age | 0.00366 *** (0.000498) | 0.101 *** (0.00242) | −0.0310 *** (0.00262) | 0.0693 *** (0.00222) | 0.102 *** (0.00312) | 0.0811 *** (0.00323) | −0.0180 *** (0.00339) |
| Bsize | −0.0100 *** (0.00257) | | | | | | |
| Bindep | 0.000265 (0.000361) | | | | | | |
| Bwomen | −0.000918 (0.000761) | | | | | | |
| CSRCommittee | 0.185 *** (0.0164) | | | | | | |
| ERRI | 0.322 *** (0.0445) | | | | | | |
| IENVPI | 0.200 *** (0.0284) | | | | | | |
| EJ | −0.0386 * (0.0232) | | | | | | |
| Munif | | −0.509 *** (0.0264) | 0.691 *** (0.0397) | −0.806 *** (0.0349) | −0.925 *** (0.0362) | −1.397 *** (0.0367) | −0.440 *** (0.0423) |
| Div | | −0.000143 *** $(5.98 \times 10^{-6})$ | $9.37 \times 10^{-5}$ *** $(4.05 \times 10^{-6})$ | 0.000107 *** $(3.66 \times 10^{-6})$ | $2.16 \times 10^{-5}$ $(5.95 \times 10^{-5})$ | $-9.18 \times 10^{-5}$ *** $(3.12 \times 10^{-5})$ | $-7.62 \times 10^{-5}$ * $(4.29 \times 10^{-5})$ |
| CGScore | | 0.000987 *** (0.000134) | 0.00302 *** (0.000126) | 0.00169 *** $(7.60 \times 10^{-5})$ | 0.00242 *** (0.000143) | 0.000754 *** $(5.92 \times 10^{-5})$ | −0.000415 *** $(9.18 \times 10^{-5})$ |
| SocialScore | | −0.0126 *** (0.000292) | 0.0175 *** (0.000391) | −0.00566 *** (0.000335) | 0.00194 *** (0.000684) | 0.000967 *** (0.000333) | −0.00524 *** (0.000531) |
| NCSRPI | | 0.0354 * (0.0195) | −0.00581 (0.0249) | −0.000929 (0.0153) | 0.000 (0.000) | 0.000 (0.000) | 0.000 (0.000) |
| Industry, Country, Year and Crisis controlled | | | | | | | |
| Loglikelihood | −8248.1466 | | | | | | |
| p-value | 0.000 | | | | | | |
| AR(1) | | −1.25 | −3.29 | −1.06 | −1.11 | −0.83 | 2.49 |
| AR(2) | | 1.09 | 0.95 | −1.03 | 0.99 | 0.05 | −1.01 |
| Hansen Test | | 404.59 | 389.41 | 356.10 | 325.83 | 295.44 | 271.75 |

### 4.3. Robustness Analysis

In order to check the robustness of the results obtained in the main analysis, Table 4 shows the results obtained using the variables that represent economic profitability (ROA) and financial profitability (ROE) as proxies of business performance. These variables are accounting figures that represent the profitability of the investments that a company has made from the point of view of the investment in assets and the shareholders, respectively. Furthermore, in this analysis some variables considered in the initial model that were not significant from the econometric point of view have been omitted.

**Table 4.** Robustness analysis for profitability (*** $p < 0.01$, ** $p < 0.05$, * $p < 0.1$).

| | Panel A. Environmental Innovations and Return on Assets | | | | | |
|---|---|---|---|---|---|---|
| | ROA | | | | | |
| | t | t + 1 | t + 2 | t + 3 | t + 4 | t + 5 |
| | Coeff. (Std. Error) | Coeff. (Std. Error) | Coeff. (Std. Error) | Coeff. (Std. Error) | Coeff. (Std. Error) | Coeff. (Std. Error) |
| $ROA_{t-1}$ | 0.00906 *** | 0.00625 ** | −0.0812 *** | 0.00809 ** | −0.0231 *** | −0.0336 *** |
| | (0.00334) | (0.00270) | (0.00281) | (0.00367) | (0.00463) | (0.00861) |
| Env_Inno | −1.771 *** | −0.00355 | −1.680 *** | −0.191 ** | −0.0114 | 0.160 * |
| | (0.0731) | (0.0630) | (0.109) | (0.0938) | (0.0437) | (0.0823) |
| Size | 2.058 *** | −6.211 *** | −1.624 *** | −1.862 *** | 1.179 *** | 1.074 *** |
| | (0.104) | (0.142) | (0.134) | (0.194) | (0.147) | (0.184) |
| Leverage | −1.42 × 10⁻⁵ *** | −3.21 × 10⁻⁵ *** | 1.34 × 10⁻⁵ *** | 2.02 × 10⁻⁵ *** | 0.000530 *** | −0.000150 *** |
| | (1.38 × 10⁻⁶) | (1.47 × 10⁻⁶) | (2.12 × 10⁻⁶) | (2.53 × 10⁻⁶) | (1.35 × 10⁻⁵) | (1.29 × 10⁻⁵) |
| Inter | 0.0213 *** | 0.0203 *** | −0.0150 *** | −0.0412 *** | 0.000114 | 0.0164 *** |
| | (0.00256) | (0.00294) | (0.00384) | (0.00524) | (0.00261) | (0.00443) |
| CAPEX | 0.0460 *** | 0.0375 *** | 0.0759 *** | 0.0174 *** | −0.0497 *** | 0.0455 *** |
| | (0.000435) | (0.000153) | (0.000471) | (0.00154) | (0.00106) | (0.00227) |
| R&D | −0.00119 *** | −0.00121 *** | −0.00270 *** | −0.0223 *** | 0.0101 *** | 0.0297 *** |
| | (3.91 × 10⁻⁵) | (1.21 × 10⁻⁵) | (0.000618) | (0.00122) | (0.000601) | (0.000616) |
| WC | 1.64 × 10⁻¹⁰ *** | −3.46 × 10⁻¹⁰ *** | 7.81 × 10⁻¹¹ *** | 4.79 × 10⁻¹⁰ *** | 2.04 × 10⁻⁹ *** | −4.40 × 10⁻¹⁰ *** |
| | (0.000) | (0.000) | (0.000) | (0.000) | (9.04 × 10⁻¹¹) | (1.07 × 10⁻¹⁰) |
| F_Age | −0.274 *** | −0.236 *** | 0.301 *** | 0.0595 | 0.197 *** | −0.205 *** |
| | (0.0277) | (0.0187) | (0.0357) | (0.0382) | (0.0340) | (0.0345) |
| Munif | 6.762 *** | −0.473 * | −2.412 *** | 7.070 *** | −0.941 ** | −5.582 *** |
| | (0.285) | (0.277) | (0.444) | (0.602) | (0.407) | (0.407) |
| Div | 0.00142 *** | 1.10 × 10⁻⁵ | −0.0113 *** | −0.00302 *** | −0.00232 *** | −0.00876 *** |
| | (9.05 × 10⁻⁵) | (4.43 × 10⁻⁵) | (9.52 × 10⁻⁵) | (0.000138) | (9.56 × 10⁻⁵) | (8.05 × 10⁻⁵) |
| CGScore | −0.00245 *** | 0.0101 *** | −0.00798 *** | 0.0355 *** | −0.00619 *** | 0.0122 *** |
| | (0.000560) | (0.00103) | (0.00126) | (0.00118) | (0.000998) | (0.00139) |
| SocialScore | −0.104 *** | 0.238 *** | −0.129 *** | −0.0447 *** | 0.0132 *** | 0.0231 *** |
| | (0.00320) | (0.00376) | (0.00398) | (0.00727) | (0.00354) | (0.00458) |
| | Industry, Country, Year and Crisis controlled | | | | | |
| AR(1) | −2.93 | −2.57 | −0.98 | −2.47 | −2.23 | −2.50 |
| AR(2) | 0.34 | −0.07 | 0.54 | 0.23 | −2.13 | −1.45 |
| Hansen Test | 340.35 | 326.45 | 313.65 | 300.31 | 265.11 | 241.24 |
| | Panel B. Environmental Innovations and Return on Equity | | | | | |
| | ROE | | | | | |
| | t | t + 1 | t + 2 | t + 3 | t + 4 | t + 5 |
| | Coeff. (Std. Error) | Coeff. (Std. Error) | Coeff. (Std. Error) | Coeff. (Std. Error) | Coeff. (Std. Error) | Coeff. (Std. Error) |
| $ROE_{t-1}$ | −0.00294 *** | −0.00925 *** | 0.00591 *** | 0.234 *** | 0.184 *** | 0.0386 *** |
| | (2.60 × 10⁻⁵) | (2.02 × 10⁻⁵) | (4.94 × 10⁻⁵) | (0.00455) | (0.00328) | (0.00107) |
| Env_Inno | −4.812 *** | −8.808 *** | −5.788 *** | −4.047 *** | −7.553 *** | 2.245 *** |
| | (0.251) | (0.239) | (0.245) | (0.349) | (0.297) | (0.212) |
| Size | 25.86 *** | −39.46 *** | −2.138 *** | −2.658 *** | −6.306 *** | 15.16 *** |
| | (0.577) | (0.728) | (0.586) | (0.693) | (0.792) | (0.481) |
| Leverage | −0.0105 *** | 0.0152 *** | 0.00168 *** | 0.00132 *** | −0.00255 *** | −0.000501 *** |
| | (0.000291) | (5.89 × 10⁻⁵) | (3.46 × 10⁻⁶) | (1.74 × 10⁻⁵) | (0.000102) | (4.29 × 10⁻⁵) |
| Inter | −0.326 *** | −0.278 *** | −0.0877 *** | 0.159 *** | 0.556 *** | 0.402 *** |
| | (0.00754) | (0.00996) | (0.0109) | (0.0153) | (0.0170) | (0.0168) |
| CAPEX | 0.0678 *** | 0.0206 *** | 0.124 *** | −0.0308 *** | −0.135 *** | −0.195 *** |
| | (0.000596) | (0.000732) | (0.00173) | (0.00185) | (0.0325) | (0.0228) |
| R&D | −0.00170 *** | −0.00100 *** | −0.00177 | 0.0332 *** | 0.531 *** | 1.077 *** |
| | (0.000140) | (9.00 × 10⁻⁵) | (0.00123) | (0.00393) | (0.00346) | (0.00255) |
| WC | 2.96 × 10⁻¹⁰ *** | 3.61 × 10⁻⁹ *** | −4.42 × 10⁻⁹ *** | 3.33 × 10⁻⁹ *** | 8.85 × 10⁻⁸ *** | 9.14 × 10⁻⁹ *** |
| | (0.000) | (7.19 × 10⁻¹¹) | (7.07 × 10⁻¹¹) | (8.25 × 10⁻¹¹) | (1.17 × 10⁻⁹) | (2.01 × 10⁻¹⁰) |
| F_Age | −1.788 *** | 3.662 *** | 1.132 *** | 3.746 *** | 3.274 *** | 0.277 *** |
| | (0.0705) | (0.100) | (0.120) | (0.177) | (0.190) | (0.105) |
| Munif | 3.371 *** | −15.32 *** | 36.91 *** | 18.96 *** | 6.277 *** | −22.49 *** |
| | −1.154 | −1.511 | −1.450 | −2.173 | −2.075 | −1.690 |
| Div | −0.00206 *** | 0.0368 *** | −0.0883 *** | −0.198 *** | 0.0905 *** | −0.273 *** |
| | (0.000233) | (0.00123) | (0.000309) | (0.00491) | (0.000974) | (0.00221) |
| CGScore | 0.0259 *** | −0.00316 | 0.0592 *** | 0.255 *** | 0.0536 *** | 0.00478 |
| | (0.00287) | (0.00340) | (0.00297) | (0.00667) | (0.00583) | (0.00516) |
| SocialScore | −0.281 *** | 0.317 *** | −0.668 *** | −0.877 *** | −0.169 *** | −0.727 *** |
| | (0.00620) | (0.0108) | (0.00903) | (0.0249) | (0.0197) | (0.0275) |
| | Industry, Country, Year and Crisis controlled | | | | | |
| AR(1) | 0.16 | 0.23 | −0.30 | −0.52 | −2.20 | −1.54 |
| AR(2) | 0.67 | 1.09 | 0.97 | 1.05 | −1.07 | −0.95 |
| Hansen Test | 347.72 | 349.40 | 334.50 | 313.15 | 259.37 | 227.41 |

In this regard, as shown in Table 4 the impact of environmental innovations on both proxies of business performance is very similar to that obtained for Tobin's Q. Specifically, the results reflected in Panel A of Table 4 show that environmental innovations lead to lower levels of economic profitability (ROA) in the year in which the investment is made and during the following four years, with a change in this trend beginning in the year t + 5, when the proactive investments in environmental issues begin to have a positive and significant impact on ROA. In Panel B of Table 4 it can be seen that this impact is more marked when the financial profitability (ROE) is considered.

## 5. Conclusions

This paper aimed to examine the role that CEO power plays in environmental innovation and the impact that these strategies have on financial performance. Specifically, we aimed to answer the following research questions: (1) How does CEO power affect eco-innovation? and (2) How does eco-innovation affect firm performance? Both issues have been the subject of considerable debate in the literature, with opposite views and contradictory findings.

Using data from a sample of 4863 international firms corresponding to the period 2002–2017, our results indicate that investing in environmental innovations related to the use of clean technologies, ecological production processes, and the design, manufacture and commercialization of environmentally sustainable products requires that CEOs have a greater level of power in order to support projects that do not entail a higher return in the short and medium terms. Thus, although some prior studies pose that CEO power causes agency problems, reduces the effectiveness of the board's monitoring role, and reduces the likelihood of approving long-term capital investments in environmental projects, leading to a decline in environmental performance (De Villiers et al. 2011; Harper and Sun 2019; Sheikh 2019), we demonstrate that the accumulation of power in the figure of the CEO can be beneficial for the firm. Specifically, our findings confirm prior findings regarding a "bright side" of CEO power with regard to corporate sustainability (Walls and Berrone 2017; Li et al. 2018; Velte 2019).

Furthermore, our findings also show that the negative economic effect of eco-innovation reverses in the fourth and fifth years after environmental innovations were implemented. Thus, our findings indicate that, at least, a period of four years is necessary for the positive effect of eco-innovation on financial performance comes into fruition. These results are robust for different specifications of firm performance, considering this magnitude both from the point of view of the stock market and the returns on assets and shareholders

Our findings have some theoretical and practical implications. From a theoretical point of view, our results confirm the arguments regarding the existence of a lag between the implementation of the eco-innovation strategy and its tangible results, in such a way that the potentially positive effects of eco-innovation on performance come into fruition at a later date (Alos-Simo et al. 2020). Moreover, our findings contribute to the debate about whether the concentration of power in the figure of the CEO may be harmful or beneficial to the firm by showing that CEO power has a positive impact on the firm's long-term value by promoting environmental innovative projects. In this sense, our findings provide support to the arguments regarding the influence of the CEO's individual characteristics on proactive environmental strategies (Arena et al. 2018; García-Sánchez et al. 2020c).

With regard to the implications for practice arising from this study, our findings have various managerial and policy implications. From a managerial viewpoint, we demonstrate that, to the extent that CEOs have a high environmental awareness, empowering the CEO represents an opportunity rather than a threat to increase firm value. Furthermore, we contribute to the discussion and understanding of the role that CEO power plays in a specific type of strategic decisions, i.e., those related to environmental innovation. In this sense, a better understanding about the impact that the accumulation of power in the figure of the CEO has on corporate strategies and policies allows companies to design adequate monitoring structures to foster environmental innovation and refine their

monetary incentives plans, linking CEO compensation plans to indicators of environmental sustainability or long-term financial performance.

Additionally, our findings indicate that, as regards the eco-innovation-performance link, the time perspective matters, since a long period of time (at least four years) must elapse before the materialization of the positive effect of eco-innovation projects on financial performance. Consequently, our findings suggest that, if firms really want to improve their competitiveness, they should avoid a "myopic focus" and take a long-term perspective, promoting such proactive environmental strategies. Furthermore, given the key role that the CEO plays in promoting eco-innovation projects, being aware that they will reap the rewards of eco-innovation in the long term means firms could mitigate CEOs' career concerns.

In regard to CEOs, demonstrating that eco-innovation projects enhance the company's financial performance in the long term, our results encourage them to be "patient" when considering these types of strategy and, to some extent, alleviate the career or reputational concerns that could lead them to adopt a "myopic viewpoint" and, consequently, under-invest in eco-innovation projects. Similarly, from the investors' perspective, these findings provide confidence that eco-innovation projects will have a positive impact on firm value and, thus, support the change of mentality in investors in relation to this type of investment documented in previous studies.

As to the policy implications of this study, firstly, the results provide confidence to policy makers regarding the positive effect that the policies that promote eco-innovation can have on business competitiveness and, secondly, offer some guidelines that could assist them to establish incentives for developing eco-innovation. Indeed, a better under-standing of the drivers of eco-innovation could assist policy makers to develop policies aimed at promoting eco-innovation (Keshminder and Río 2019; Costa 2021), for example, stimulating or, at least, not limiting the accumulation of power in the figure of the CEO. Moreover, considering this paper's results, policy makers could develop policies aimed at raising managerial awareness regarding the environmental and economic benefits that eco-innovation projects can generate.

Finally, it should be noticed that this paper is subject to some limitations. Following Garcia-Sanchez et al. (2020d), we measured CEO power as an index that reflects three dimensions related to CEO structural power (i.e., if the CEO is a member of the board of directors, if she/he exercises the functions of chairman, and if the percentage of executive directors on the board is higher than the average). However, we are aware that CEO power can stem from several sources (i.e., structural, ownership, expert, and prestige) and, consequently, can be measured by using different proxies (e.g., CEO-chair duality, CEO share ownership, CEO pay slice, CEO tenure, CEO connectedness, CEO founder status). In this sense, future studies could complete the analysis by employing alternative proxies of CEO power. Furthermore, although we considered a broad set of control variables, there are other potential variables (e.g., having the CSR report assured or ESG score) that could be included in the models. Future studies could consider analyzing the effect of these new variables. Finally, further analysis is required with regard to the relationship between the control variables considered in this study and Tobin's Q considering different lags.

**Author Contributions:** The whole article is the result of a joint project and shared effort: Conceptual-ization, B.A.-G. and J.-V.F.-A.; methodology, B.A.-G. and J.-V.F.-A.; validation, B.A.-G. and J.-V.F.-A.; formal analysis, B.A.-G. and J.-V.F.-A.; investigation, B.A.-G. and J.-V.F.-A.; resources, B.A.-G. and J.-V.F.-A.; data curation, B.A.-G. and J.-V.F.-A.; writing—original draft preparation, B.A.-G. and J.-V.F.-A.; writing—review and editing, B.A.-G. and J.-V.F.-A.; visualization, B.A.-G. and J.-V.F.-A.; supervision, B.A.-G. and J.-V.F.-A.; project administration, B.A.-G. and J.-V.F.-A. Both authors have read and agreed to the published version of the manuscript.

**Funding:** This research received no external funding.

**Institutional Review Board Statement:** Not applicable.

**Informed Consent Statement:** Not applicable.

**Data Availability Statement:** The data presented in this study are available on request from the corresponding author.

**Conflicts of Interest:** The authors declare no conflict of interest.

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
