# Peer review of "Is It Necessary to Centralize Power in the CEO to Ensure Environmental Innovation?"

_admsci, doi:10.3390/admsci11010027_

Round 1
Reviewer 1 Report
This paper is very interesting and very well written. However, I have several concerns as explained next. Hope you find them useful to improve your research. Good luck!
Major concerns:
- In H1, CEO power is not well justified. It seems strange that having more directors in the board than the average means more power to the CEO. Note that Chen 2014 gives other focus which seems to be more meaningful based on four indicators of CEO power: CEO duality (i.e., whether the CEO is also the board chair), the ratio of shares held by the CEO to director ownership, the ratio of non‐independent directors to the total number of directors, and the ratio of directors who were appointed after the CEO began his or her tenure to the total number of directors. Please consider if you can take some of these items in your proxy for CEO power.
- The methodological approach is not clear. Why use GMM in eq2 but not in eq1? The table 3 lacks important information such as probability values to validate GMM (AR2 and Hansen test).
- There are in my view too many control variables which are not sufficiently explained (e.g. expected signs according to literature), and which are not the same in eq. 1&2 which is somehow confusing.
Minor points:
Abstract: "is reversed from the fourth after proactive environmental innovations are promoted"--> fourth year??. Be consistent as in l. 68 you say "fourth and fifth year"
l. 146 without having to face the possible opposition of the board to this type of projects (García-Sánchez et al., 2020c) that do affect less powerful CEOs inhibiting eco-innovation.--> be careful when you cite. According to Garcia-Sanchez et al 2020c, it is the board of directors the body responsible for firm strategy. They analyse the role of directors but do not explore how powerful the CEO is and the impact of that power.
l.225: explain here more clearly how you measure CEOpower and specify the control variables. Use citations to help the reader understand what to expect according to prior literature. Explain controls and crisis also in eq 2, citing existing research and commenting on the expected signs.
Give literature to back up company-specific effect, η. I cannot clearly see why it is needed, as you are already controlling for the characteristics of the firm, etc...
What years do you consider as crisis years? Can you justify that? Should not you eliminate "year" if "crisis" is introduced?- if crisis is dependent on some years, you seem to be duplicating the factor you are controlling for.
Is “Env_Inno” and index from 0-3 depending on the 3 items (1 point each?) you explain in l.240-242? Please clarify.
Explain sectorial munificience level and the sign you expect- as for the other control variables if clearly stated and confirmed by previous studies.
Why do you include Div, CAPEX, WC etc. as control variables and do not include important control variables as having the CSR assured, or ESG score (instead of GCscore or social score yet not the environmental score)?
It is odd how the sign of the control variables change from + to - if you change the lag in Tobin Q- see for instance Inter or F-age which you highlight in your results in l 324 .
It is also very odd that you do not keep the same control variables in both equations.
l. 316 prior research report that-->prior research reports that
Translate into English the words in Spanish that appear in the keyword, tables, etc.
Why do you not use GMM for eq 1? You need to report probability for Hansen test and AR2 as in Martinez-Ferrero & Garcia-Sanchez (2017), Garcia-Sanchez & Martinez-Ferrero (2018).
Very highly cited studies pose that the CEO-Chair duality causes agency problem, reduces the effectiveness of monitoring role of the board, and reduces the likelihood of approving long-term capital investment in environmental projects, leading to a decline in environmental performance (de Villiers et al, 2011, for example). I think you should acknowledge that you try to demonstrate the opposite and also comment on other papers coincident with your resultsto back up your findings and conclusions.
Table 3 is missing some *** in many cells. Please check.
Revise format of references. Some of them are missing the italics or are completely in italics.
References:
De Villiers, C., Naiker, V., & Van Staden, C. J. (2011). The effect of board characteristics on firm environmental performance. Journal of Management, 37(6), 1636-1663.
Martínez‐Ferrero, J., & García‐Sánchez, I. M. (2017). Sustainability assurance and cost of capital: Does assurance impact on credibility of corporate social responsibility information?. Business Ethics: A European Review, 26(3), 223-239.
García‐Sánchez, I. M., & Martínez‐Ferrero, J. (2018). How do independent directors behave with respect to sustainability disclosure?. Corporate Social Responsibility and Environmental Management, 25(4), 609-627.
Garcia-Sanchez, I. (2020). Corporate social reporting and assurance: The state of the art. Spanish Accounting Review/Revista de Contabilidad.
Author Response
Dear Reviewer 1,
We are extremely grateful with your detailed comments and helpful suggestions. We believe that they allowed us to improve the relevance and quality of our paper and we have considered all of them. If you consider that our interpretation of some of your comments is wrong due to linguistic differences, sorry for the confusion.
We have reinforced the manuscript, special those issues related to your major concerns as well as most to the minor points that you indicated:
- Firstly, we have tried to better justify hypothesis H1 regarding the relationship between CEO power and eco-innovation. In this regard, based on your comments, we have acknowledged that prior studies pose that CEO power causes agency problem, reduces the effectiveness of monitoring role of the board, and reduces the likelihood of approving long-term capital investment in environmental projects, leading to a decline in environmental performance, and we have stressed that we posit the opposite indicating some papers that share our view. However, we must clarify that we have maintained the initial way of measuring CEO power - i.e., the measure proposed by García-Sánchez et al. (2020d) - due to the limited time to introduce such a major modification. Nevertheless, we have recognized the existence of different proxies for CEO power, perhaps more representative than ours, and we have incorporated their use as possible future lines of our study.
- Secondly, based on your comments, we have introduced several changes in section 3 in order to clarify the methodological approach of the study. In the first place, we have restructured the section, creating a sub-section intended to explain the analysis techniques used in each case (we believe that some of your comments were the result of an unclear wording of the methodology used). We also explained carefully all variables (both their measurement and their meaning) by using citations to help the reader understand what to expect according to prior literature. We hope we have answered all the questions you raised in relation to the different variables. However, in this respect, although we agree with you about CSR assurance or ESG score are important control variables, we did not include them in the models, due to the limited time to introduce such a major modification. Nevertheless, we have recognized this possibility as a future extension of our study.
- Thirdly, we structured the findings around the research hypotheses, including references to prior studies that have obtained similar/different results.
- Fourthly, we eliminated or rewrote some confusing sentences related to the opposition of the board. We also rewrote the abstract.
- Finally, we have revised the format of the references and we have corrected the phrases that you have pointed out in your review and we used a professional proofreading service to be sure to edit the manuscript with a perfect level of academic English.
Thanks for your time and effort.
Kind Regards,
The authors
Reviewer 2 Report
The article is well accomplished with a good methodological approach. Moreover, the authors were very careful in the introduction, highlighting the research gap, the objectives and the contributions of the article. The literature review is based on relevant and current articles. For these reasons the article is a good candidate for publication after the resolution of the following weakness:
- The discussion and conclusions fall far short of the rest of the article. As I mentioned, the article is very careful, but this part needs more investment, separating the conclusions from the discussion and including the practical/managerial implications.
Author Response
Dear Reviewer 2,
We are extremely grateful with your detailed comments and helpful suggestions. We believe that they allowed us to improve the relevance and quality of our paper and we have considered all of them.
We have reinforced the manuscript, special those issues related to the discussion of the findings and the conclusions. Thus, we have tried to better explain the study’s findings. Specifically, we structured the findings around the research hypotheses, including references to prior studies that have obtained similar/different results. Additionally, we reinforce the conclusions stressing the main managerial and practical implications of our findings and including some future research avenues. Finally, we have used a professional proofreading service to be sure to edit the manuscript with a perfect level of academic English.
Thanks for your time and effort.
Kind Regards,
The authors
Reviewer 3 Report
At first I would like to congratulate the authors for the work already made and thank the opportunity of reading their work and contribute with comments that I believe will contribute for the improvement of the article.
The literature connecting the CEO profile with eco-innovative strategies is overlooked, as a consequence I appreciate the effort made to fulfill this gap.
In what relates to the present version I have some points to which I feel more attention should be paid:
- At first I would like the authors to reconsider the underlying reasoning for the affirmation ".....promote investments in projects that in the short and medium term are negatively valued by the capital market (line 10) as I do not believe that the capital market negatively values these investments.
- Along the document some sentences are very hard to follow, so, I strongly recommend the authors to re-read and correct them as well as making the document go through a professional proof reading. (e.g "Additionally, results indicate that the negative impact of eco-innovation on financial performance is reversed from the fourth after proactive environmental innovations are promoted, so that their effects on financial performance are positive in the long term."
- As a major concern, I highlight the importance of debating in addition to the CEO power there needs to be a debate on the CEO mindset, as the real hinderer can be her unwillingness to eco-innovate. This needs to be considered.
- Another major aspect that has to be debated is the role played by the public policy towards firm strategic behaviour, in this vein, I found a very recent reference that could be of help: Costa, J. Carrots or Sticks: Which Policies Matter the Most in Sustainable Resource Management? Resources 2021, 10, 12. https://doi.org/10.3390/resources10020012 The CEO is bounded by extant regulations and that is an aspect which deserves being addressed.
- Concerning hypothesis 2, I feel that some adjustment/reinforcement is required, as all innovative actions are risky by nature and produce delayed outcomes.
- There is a need for underlying the singularities of eco-innovations as this type of innovations correct market failures, which gives them a social dimension rather than an individual dimension.
- Concerning the empirical procedures there are also some important hesitations: the simultaneous inclusion of country and year controls - the authors refer the existence of 70 countries and the inclusion of 16 years, which with simplistic calculations puts nearly 38 observations (on average) leading to potentially unfeasible estimations. I do not understand the importance of country controls with so much diversity. I believe that other segmentation needs to be considered.
- Also equation 2 proposes a Model to which I have hesitations: why is TOBIN's q lagged? Panel data estimations guarantee, per se the control for endogeneity. In my view, if the authors propose a dynamic panel there should be a lagged term for eco innovation as the effects of the investment are not immediate.
- Still in relation to the second equation there are variables which are time invariant (such as the country and the industry) - this estimation only becomes feasible with the Wooldridge correction (1995).
- The introduction of a variable "crisis", is in my view dangerous as the financial crisis has hit asymmetrically the countries. Moreover, that is not the major point of the article - it should be removed or included as a structural break.
- The descriptives should encompass min and max.
- The correlation table presents some awkward results as too much of the bivariate correlations are 0 or close. that does not seem accurate. please revise.
- In table 3 I do not understand the forward looking estimation. Please enlight the reader of this purpose. There should be included 3 dp as there is no interest in so many dp not even in scientific notation.
- There is a need for a more solid debate on the results. But, there is even more to be added on the connection to extant literature.
- There is a need to add a section on theoretical and practical implications as well as policy recommendations.
Thanks for considering these remarks,
Best luck
Author Response
Dear Reviewer 3,
We are extremely grateful with your detailed comments and helpful suggestions. We believe that they allowed us to improve the relevance and quality of our paper and we have considered all of them. If you consider that our interpretation of some of your comments is wrong due to linguistic differences, sorry for the confusion.
We have reinforced the manuscript, special those issues related to your major concerns:
- Firstly, we have restructured the theoretical framework in order to connect the CEO profile with eco-innovative strategies. We introduced a sub-section intended to explain the singularities of eco-innovations and the role of managerial concern. This allows us to highlight how the CEO’s mindset affects her/his willingness to eco-innovate and, consequently, we think that we were able to better justify hypothesis H1 regarding the relationship between CEO power and eco-innovation. In this regard, we have included references to some papers that share our view. Additionally, following your comments, we reinforced the development of hypothesis H2 (considering both the case of general innovation and the singularities that characterize eco-innovations). Again, we support our arguments with prior studies’ findings.
- Secondly, based on your comments, we have introduced several changes in section 3 in order to clarify the methodological approach of the study. In the first place, we have restructured the section, creating a sub-section intended to explain the analysis techniques used in each case (we believe that some of your comments were the result of an unclear wording of the methodology used). We also explained carefully all variables (both their measurement and their meaning) by using citations to help the reader understand what to expect according to prior literature. We hope we have answered all the questions you raised in relation to the different variables.
- Thirdly, we structured the findings around the research hypotheses, including references to prior studies that have obtained similar/different results.
- Fourthly, we reinforce the conclusions stressing the main managerial and practical implications of our findings and including policy recommendations as well as some future research avenues.
- Fifthly, we eliminated or rewrote some confusing sentences related to how the capital market values eco-innovations. We also rewrote the abstract.
- Finally, we have corrected the phrases that you have pointed out in your review and we used a professional proofreading service to be sure to edit the manuscript with a perfect level of academic English.
Thanks for your time and effort.
Kind Regards,
The authors
Reviewer 4 Report
- Prior research and differentiation problem due to lack of logical connection
① Why do companies with CEO power invest in environmental innovation? Rather than the logical basis for the content to be asserted, only previous studies are listed. First, if a company wants to invest in environmental innovation, it is argued that the CEO must have greater power. Why? There must be a logical basis for what you want to argue in this study, and the contents of Hypothesis 1 have already been examined in previous studies. This study enumerates studies on CEO power and firm innovation, and then examines the relationship between CEO power and environmental innovation without a logical basis. It is difficult to understand logically.
Simple innovation and environmental innovation have different meanings. If a CEO who cares about the environment would be interested in environmental innovation, if not, would it not be more interested in the executives' pursuit of private interests if there is CEO power? The rationale for why we are more concerned about environmental innovation when there is CEO power must be described in the study.
② There are some previous studies showing that environmental innovation has a positive effect on corporate value and there are cases where it does not. The impact of environmental innovation on corporate value has already been done in other previous studies, so it seems that there is no difference from previous studies. The reviewer thinks that there is a point that is insufficient to differentiate it from previous studies by using corporate value as a long-term variable.
The contents of Hypothesis 1 and Hypothesis 2 have already been dealt with in previous studies, so there is not much difference and contribution from previous studies.
Author Response
Dear Reviewer 4,
We sincerely regret your opinion, although we do not agree. We do believe that our study is an original work with clear theoretical and practical contributions. We also believe that both hypotheses are interesting, since they address two "hot topics" on which there are no conclusive results.
However, we have reinforced the manuscript, special those issues related to the development of hypothesis, methodology and the discussion of the findings and the conclusions. Thus, we have tried to better explain the study’s findings, including references to prior studies that have obtained similar/different results. Additionally, we reinforce the conclusions stressing the main managerial and practical implications of our findings and including some future research avenues. Finally, we have used a professional proofreading service to be sure to edit the manuscript with a perfect level of academic English.
Thanks for your time and effort.
Kind Regards,
The authors
Round 2
Reviewer 2 Report
Congratulations on this revision. Nice work.
Author Response
Dear Reviewer 2,
We want to thank you for the favorable opinion of the second reviewer.
Thanks for your time and effort.
Reviewer 3 Report
Many thanks to the authors for providing an improved version of the article. The document has importantly improved, however some additional efforts are still required for publication.
- At first the introduction should be re-written as the paper objective should be more emphasized rather than the database in use. The implications of the findings must be mentioned too.
- The referencing styles still does not match the requests of the journal - please adjust.
- There are still some words in Castellano in the tables - please correct.
- I would like the debate to bring more light other performance indicators such as ROA and ROE (they are also in the equation) as Tobin's Q can be a debatable choice given the sectoral heterogeneity and the presence of service sectors.
- As the database is not widely known, there is a need for some descriptive section to characterize the database selection. Then the results will become more understandable.
- There are too much variables in the regression - please reconsider the removal of some of them - from the correlation table you can see that they barely relate to the others.
- The correlation table needs to be rotated to become understandable - it cannot be a multiple page table.
- Table 3 does not look as expected - avoid scientific notation and use 3dp.
- There is a need for more policy implications debate, also what are the managerial implications.
- in sum, the paper must end making it clear to the reader about the worth of your work. Which questions are answered? which questions remain unanswered? What can be learned for the future?
Congratulations for what you have done so far, and consider these final efforts to make your paper more solid.
All the best
Author Response
Title: IS IT NECESSARY TO CENTRALIZE POWER IN THE CEO TO ENSURE ENVI-RONMENTAL INNOVATION?
Dear Reviewer 3,
We are extremely grateful with your detailed comments and helpful suggestions. We believe that they allowed us to improve the relevance and quality of our paper and we have considered all of them. If you consider that our interpretation of some of your comments is wrong due to linguistic differences, sorry for the confusion.
We have reinforced further the manuscript, special those issues related to your concerns:
- In the Introduction section we emphasized the objective of the paper and mentioned the main implications of the findings.
- We matched the referencing styles with the requests of the journal.
- We reviewed and corrected the Tables.
- We included a new sub-section (4.3) with a robustness analysis considering other performance indicators (i.e., ROA and ROE)
- The database selection was explained further.
- Policy and managerial implications were discussed to a greater extent.
- We made an effort to clarify our study’s contributions (i.e., the worth of our work and which questions were answered)
- Finally, we have corrected some words and phrases.
There are some formatting problems that you pointed out in your report which are due to the template used and we assume they will be resolved in the editing tests, in the event that the paper is finally accepted.
Thanks for your time and effort.
Kind Regards,
The authors
Reviewer 4 Report
This paper is well revised.
Author Response

(The authors gave the same response as above.)

Round 3
Reviewer 3 Report
Many thanks for taking the time to reformulate your paper. I feel that this version is far more solid than the first.
Best of luck.